# Nutrient sensing pathways regulating adult reproductive diapause in *C. elegans*

**Moriah Eustice**[1,¤a]☯, **Daniel Konzman**[1,2]☯, **Jeff M. Reece**[3], **Salil Ghosh**[4], **Jhullian Alston**[1,¤b], **Tyler Hansen**[5,¤c], **Andy Golden**[5], **Michelle R. Bond**[1], **Lara K. Abramowitz**[1], **John A. Hanover**[1]*

1 Laboratory of Cell Biochemistry and Biology, National Institute of Diabetes and Digestive and Kidney Diseases, National Institutes of Health, Bethesda, Maryland, United States America, 2 Johns Hopkins University Department of Biology, Baltimore, MD, United States America, 3 Advanced Light Microscopy and Image Analysis Core, National Institutes of Health, Bethesda, Maryland, United States America, 4 Center for Biologics Evaluation and Research, Food and Drug Administration, Silver Spring, Maryland, United States America, 5 Laboratory of Biochemistry and Genetics, National Institute of Diabetes and Digestive and Kidney Diseases, National Institutes of Health, Bethesda, Maryland, United States America

☯ These authors contributed equally to this work.
¤a Current address: Center for Tobacco Products, Food and Drug Administration, Silver Spring, Maryland, United States America
¤b Current address: Department of Biochemistry and Molecular biophysics, Center for Science and Engineering of Living Cells, Washington University in St. Louis, St. Louis, Missouri, United States America
¤c Current address: Department of Biochemistry, Vanderbilt University School of Medicine, Nashville, Tennese, United States America
* jah@helix.nih.gov

**Data Availability Statement:** All relevant data are within the paper and its Supporting Information files.

## Abstract

Genetic and environmental manipulations, such as dietary restriction, can improve both health span and lifespan in a wide range of organisms, including humans. Changes in nutrient intake trigger often overlapping metabolic pathways that can generate distinct or even opposite outputs depending on several factors, such as when dietary restriction occurs in the lifecycle of the organism or the nature of the changes in nutrients. Due to the complexity of metabolic pathways and the diversity in outputs, the underlying mechanisms regulating diet-associated pro-longevity are not yet well understood. Adult reproductive diapause (ARD) in the model organism *Caenorhabditis elegans* is a dietary restriction model that is associated with lengthened lifespan and reproductive potential. To explore the metabolic pathways regulating ARD in greater depth, we performed a candidate-based genetic screen analyzing select nutrient-sensing pathways to determine their contribution to the regulation of ARD. Focusing on the three phases of ARD (initiation, maintenance, and recovery), we found that ARD initiation is regulated by fatty acid metabolism, sirtuins, AMPK, and the *O*-linked N-acetyl glucosamine (*O*-GlcNAc) pathway. Although ARD maintenance was not significantly influenced by the nutrient sensors in our screen, we found that ARD recovery was modulated by energy sensing, stress response, insulin-like signaling, and the TOR pathway. Further investigation of downstream targets of NHR-49 suggest the transcription factor influences ARD initiation through the fatty acid β-oxidation pathway. Consistent with these findings, our analysis revealed a change in levels of neutral lipids associated with ARD entry defects. Our findings identify conserved genetic pathways required for ARD

**Funding:** This work was supported by the Intramural Research Program of the National Institutes of Health, National Institute of Diabetes and Digestive and Kidney Diseases (JAH and AG). The funders had no role in study design, data collection and analysis, decision to publish, or preparation of the manuscript.

**Competing interests:** The authors declare no competing interests. This article was prepared while MRB was employed at NIDDK, NIH. The opinions expressed in this article are the author's own and do not reflect the views of the National Institute of General Medical Sciences.

entry and recovery and uncover genetic interactions that provide insight into the role of OGT and OGA.

## Introduction

Nutrition plays a vital role in health, development, lifespan, and most notably, health span. Dietary restriction has been found to increase longevity and improve overall markers of health [1, 2]. Specific changes in nutrient intake and the subsequent alteration to key metabolic processes is thought to contribute to the improvements in health and lifespan observed in a plethora of organisms [3, 4]. However, identifying the underlying mechanisms and their role in longevity remains an intense and controversial area of research [5, 6]. While different forms of dietary restriction can trigger divergent pathways with distinct outcomes [7], they often involve overlapping nutrient-sensing factors and changes to fat and carbohydrate metabolism [8–11].

Nutrient-sensing pathways are highly conserved, suggesting research in model organisms, such as the nematode *Caenorhabditis elegans (C. elegans)*, can inform dietary interventions and therapeutic targets relevant for improving human health span. For any organism, there is a delicate balance between nutrient acquisition, development, aging, and reproduction to ensure the health and propagation of the species. *C. elegans* is an exceptional model that has been used extensively to examine the intersection of longevity and these biological processes because it has a short life cycle, is highly fecund, is genetically amendable, and has evolutionarily conserved signaling pathways.

Changes in nutrient status affect the life cycle of *C. elegans* at several key points during development. Extended periods with limited or no nutrients can result in a diapause state, after which the animal has a full or extended lifespan upon re-feeding [12]. Adult reproductive diapause (ARD) is a state that delays reproduction and increases adult lifespan characterized by extensive germline shrinkage as worms enter adulthood while retaining one or two embryos throughout the starvation period [13]. When nematodes are starved in the mid-L4 larval stage, approximately one third of worms arrest in the L4 stage, one third undergo bagging (facultative vivipary), and one third enter the ARD state [13]. In ARD, the germline shrinks to a small pool of germ cells which are capable of regenerating the entire germline after re-feeding and exit from ARD [13]. Some evidence suggests germline shrinkage in response to starvation is not exclusive to ARD, but rather occurs in all starved L4s and worms which ultimately die due to bagging [14]. More recent findings also suggest that the transgenerational sterility observed in *prg-1*/Piwi mutants defective in piRNAs may represent a dynamic form of ARD [15]. Thus, ARD is a powerful system to analyze the interplay between dietary restriction, nutrient-sensing, germline dynamics, and lifespan extension.

A number of nutrient sensors have been found to play important roles under various dietary restriction conditions, including key factors in the target of rapamycin (TOR) pathway, AMP-activated protein kinase (AMPK) pathway, insulin signaling, hexosamine signaling, and fat metabolism pathway. Teasing apart the regulation of these pathways has proven to be complicated, due in part to the complex web of interactions inherent in metabolic processes. For example, the transcription factor DAF-16/FoxO is regulated by insulin-like signaling, the TOR pathway, and sirtuin activity. In turn, DAF-16/FoxO regulates the TOR, AMPK, and many other pathways [9, 16, 17]. Numerous proteins in nutrient-sensing pathways, including components of the insulin-like signaling pathway [18–20] and AMPK [21] are post-

transcriptionally modified with *O*-linked β-*N*-acetylglucosamine (*O*-GlcNAc). The *O*-GlcNAc transferase (OGT-1/OGT) is a nutrient-sensing enzyme responsible for modifying nucleocytoplasmic proteins with *O*-GlcNAc, which can be dynamically removed by the *O*-GlcNAcase (OGA-1/OGA). OGT utilizes the endproduct of the hexosamine pathway, UDP-GlcNAc, to add this sugar onto target proteins. Previous studies have implicated this process in diapause states and longevity in *C. elegans* [22–25]. Further, other nutrient-sensing cellular components also interact physically and/or genetically with OGT and OGA [26, 27]. While OGT and OGA are at the hub of many metabolic pathways they have not yet been analyzed extensively for their genetic interactions and functions in longevity, in part due to their being essential in most organisms. However, *ogt-1* and *oga-1* loss of function mutants are viable in *C. elegans* [22, 23], offering a unique opportunity to explore genetic interactions and analyze their role in ARD.

Here, we explored critical nutrient-sensing pathways to determine which genes are essential for entry, maintence, and exit from ARD, and what metabolic changes occur. Using a candidate gene approach, we examined key nodes in fat metabolism, insulin signaling, TOR, AMPK, sirtuin, and *O*-GlcNAc cycling pathways. These results have significant implications for understanding the dynamic activity of nutrient sensors in diapause, longevity, and the important functions of *O*-GlcNAc cycling.

## Materials and methods

### Strains

*C. elegans* strains were maintained and cultured under standard laboratory conditions at 20˚C with the food source *E. coli* OP50 as previously described [28]. Specific alleles of *ogt-1* and *oga-1* used in this study have previously been characterized, including *ogt-1(ok1474)*, *ogt-1(ok430)* [22], *oga-1(tm3642)* [29], and *oga-1(ok1207)* [23]. The alleles *ogt-1(jah01)*, *oga-1(av82)*, and *oga-1(av83)* were generated during the course of this study using CRISPR/ Cas9, as described below. The strain *daf-16(mu86)* was a gift from Dr. Michael Krause. We are grateful to Dr. Mark Van Gilst and Dr. Todd Lamitina for kindly gifting us the *nhr-49 (nr2041) allele and the OG1139 ogt-1(dr84[ogt-1*::*GFP] dr89[K957M]) strain, respectively*. The strains RB754 *aak-2(ok524)*, RB1988 *acs-2(ok2457)*, TJ1052 *age-1(hx546)*, BX153 *fat-7 (wa36)*, RB1206 *rsks-1(ok1255)*, VC199 *sir-2.1(ok434)*, and QV225 *skn-1(zj15)* were obtained from the *Caenorhabditis* Genetics Center (CGC, University of Minnesota, Minneapolis, MN) strain bank, which is funded by the NIH Office of Research Infrastructure Programs (P40 OD010440). All of these strains were backcrossed to our wild-type N2 Bristol laboratory strain a minimum of four times prior to analysis to ensure homogenous genetic background.

Genotyping was conducted using nested PCR, as described by the CGC, or using sequencing (Eurofins Genomics, Louisville, KY) in the case of point mutants. Primers for *ogt-1 (ok1474)*, *ogt-1(ok430)*, *oga-1(tm3642)*, and *oga-1(ok1207)* are described elsewhere [23, 29] and are available upon request. Primers for *aak-2(ok524)*, *acs-2(ok2457)*, *rsks-1(ok1255)*, and *sir-2.1(ok434)* were derived from the sequences available on wormbase.org and CGC. Primers for *daf-16(mu86)*, *nhr-49(nr2041), ogt-1(dr84[ogt-1*::*GFP] dr89[K957M])*, and the point mutants *age-1(hx546)*, *fat-7(wa36)*, and *skn-1(zj15)* were generated using Primer3 (Rozen and Skaletsky 2000) and are available on request.

Double mutants were generated by crossing, with the presence of deletion alleles verified either using nested PCR or by sequencing, in the case of point mutants. Double mutants were backcrossed a minimum of four times to our wild-type N2 Bristol strain before use in analysis.

## Generation of CRISPR/Cas9 whole gene knockouts of *ogt-1* and *oga-1*: *Ogt-1(jah01)*, *oga-1(av82)*, and *oga-1(av83)*

The alleles *ogt-1(jah01)*, *oga-1(av82)*, and *oga-1(av83)* were generated during the course of this study using CRISPR/Cas9 genome editing. These alleles are full gene deletions of the ORF of *ogt-1* and *oga-1*, respectively. We used the direct-delivery protocol developed by the Seydoux lab [30] and screened for edits using the co-conversion *dpy-10* method [31]. The crRNAs and repair template sequences are listed below. The repair template contains the deletion and perturbs the PAM sites to prevent Cas9 from re-cutting. From 15 injected P0s, we identified 3 "jackpots" and picked a total of 40 rollers from them. Using PCR genotyping, we identified and isolated three heterozygous lines from each. Homozygous animals were identified in the F2 generation and subjected to sequencing of the relevant *ogt-1* or *oga-1* region. For *ogt-1*, two lines contained the expected full deletion, which we subsequently named *ogt-1(jah01)*. With *oga-1*, two of the three lines contained the expected deletion, while the third contained an additional ~17bp deletion. The two expected *oga-1* deletions were named *oga-1(av82)*, which were the strains used in our analysis. We also kept the larger *oga-1* deletion and named it *oga-1 (av83)*.

*ogt-1* sequences:

Targeted gene crRNA1 (*ogt-1* deletion N-terminal)

`5'-AAUUUCAGAAUUAUAUCGAAGUUUUAGAGCUAUGCUGUUUUG- 3'`

Targeted gene crRNA2 (*ogt-1* deletion C-terminal)

`5'-GCUUGUGAAUAGAUUUUCGAGUUUUAGAGCUAUGCUGUUUUG- 3'`

Repair template:

`5'-Ccaatagttcaatttccaaatttcagaattatatcacacggcttgtgaatagatttt cgaaggatttta- 3'`

*oga-1* sequences:

crRNA1 (*oga-1* deletion N-terminal):

`5'-gttcacagtcaatacttaat- 3'`

crRNA2 (*oga-1* deletion C-terminal:

`5'-caaaatcgttgaatttgact- 3'`

Repair template:

`5'-ttttattaagagtgaaagttgatgacaacatttcctgattcaaattcaacgattttg ttccaaccgggatat- 3'`

## Adult reproductive diapause

Adult reproductive diapause experiments were performed as described previously [13], with minor adjustments. The overall experimental approach for initiation of ARD is illustrated in S1 Fig. Briefly, nematodes were cultured at 20˚C on nematode growth medium (NGM) with *E. coli* OP50 as the food source at a density of 3,000 per 60 mm tissue culture dish. Worms were synchronized using hypochlorite bleaching of gravid worms with a dilute alkaline hypochlorite solution, as previously described [32]. Eggs harvested from bleaching of gravid worms were then left with gentle rocking overnight at room temperature in M9 solution. Hatched L1s were then plated on NGM plus OP50 plates to complete synchronization. This synchronization process was repeated three times.

Following the third synchronization, 10,000 L1 worms (plate counting visually by sectors) were grown on a single 100 mm NGM + OP50 plate and visually inspected at intervals to ensure that over 90% of worms were in mid-L4 (approximately 48 hr, depending on growth of strain). Mid-L4 worms were then washed off the plate with M9 and collected in a 15 ml conical tube. The tubes were spun down at 20˚C for 2 min at 1500 rpm. This washing was repeated 5

times with 5 ml M9 to ensure all bacteria was removed. Prior to the fifth and final spin, worms were incubated in 5 ml M9 with gentle rocking at room temperature for 30 min to remove bacteria from gut. Following the 30 min incubation, all media was aspirated and the nematodes were suspended in 1 ml of M9, counted, and plated on high agarose tissue culture plates without bacteria/food source (ARD plates). ARD plates were prepared and poured as follows: 1 L: 3 g NaCl and 15 g agarose autoclaved in 1 L water. After autoclave sterilization, filter sterilized solutions of the following were added: 1 ml of 1M $CaCl_2$, 1 ml of 1M $MgSO_4$, 25 ml potassium phosphate buffer (pH 6.0), and 1 ml cholesterol (8 mg/ml) [13]. Using this solution, 60 mm plates were filled to two-thirds full in the hood to ensure sterility. After plates were cooled and equalized to 20˚C worms were added at a density of 3,000 worms per 60 mm plate. As a control, 3,000 of the synchronized and washed mid-L4s were plated separately from the worms that were put on the ARD plates and were instead plated on NGM plates with OP50 to allow analysis of lifespan and brood size in parallel to starved worms used to assess the ARD phenotype.

Worms were visually inspected every 24 hours (up to 72 hours, depending on strain) after plating and characterized as either in ARD, in L4, or bagged. At day 5, day 10, and day 30 after mid-L4 worms were plated on ARD plates, worms were subjected to Oil Red O (ORO) staining, carminic acid staining, lifespan analysis, and brood size counts as described below. Each experiment was repeated a minimum of three times from start to finish.

## Lifespan, germline, and brood analysis

At least 10 worms from each ARD plate were collected individually at day 5, day 10, and day 30 post-plating on ARD plates. The worms were visually inspected to ensure ARD status and placed singly onto individual NGM plus OP50 plates and incubated at 20˚C to monitor for restoration of the germline, analysis of overall number of progeny following ARD (brood size), and post-ARD lifespan. Restoration of the germline was followed visually at day 5, day 10, and day 30 using standard microscopy techniques on a Zeiss LSM 700 confocal microscope (Carl Zeiss Microscopy, LLC, Thornwood, NY) with a Plan-Apochromat 20x/0.8 objective lens. Brood size was monitored by carefully transferring the individually recovered worms every 24 hr to fresh NGM plus OP50 plates until the end of their reproductive phase, marked by the absence of eggs. Progeny of the post-ARD worms were counted after each transfer and added to total brood size reported for each animal. For all experiments, any ARD worm that failed to thrive in the first 48 hr or disappeared was censored.

The post-ARD (recovered) worms were then inspected daily after the reproductive period to determine lifespan. When worms no longer responded to gentle poking with a platinum pick, they were scored as dead. Lifespan counts for post-ARD animals includes the days in ARD, such that the lifespan post-re-feeding was added to the number of days the animals were starved. Lifespan was graphed, and statistics were performed as previously described using GraphPad and one-way ANOVA [29]. A minimum of 40 ARD animals from four independent experiments were used for analysis of each strain.

The controls also underwent the same analysis but without ever being placed on the ARD plates and, instead, were grown on standard NGM plus OP50 plates at 20˚C.

## Oil red O staining

The protocol for Oil Red O (ORO) staining was adapted from previously described methods [33] with minor adjustments. Prior to starting experiment, a stock solution of 0.5 g/100 ml ORO in isopropanol was allowed to equilibrate for at least two days at room temperature with gentle rocking and light protection. On the day of the experiment, a fresh dilution of ORO

stock solution was diluted to 60% with water and rocked for at least 1 hr. Dilute ORO was then filter sterilized with a 0.22 μm filter.

Nematodes were washed off NGM plates with OP50 (control) or high agarose ARD plates with 1 ml of 1x PBS + 0.01% Triton X-100 (PBST). Worms were washed 3 times with 1 ml of PBST and allowed to settle by gravity after each washing. Cuticles were permeabilized by resuspending worms in 120 μl of PBST and 120 μl of 2x MRWB buffer (160 mM KCl, 40 mM NaCl, 14 mM Na$_2$EGTA, 1 mM spermidine-HCl, 0.4 mM spermine, 30 mM Na-PIPES pH 7.4, and 0.2% β-mercaptoethanol) with 2% paraformaldehyde. Worms were gently rocked for 1 hr at room temperature. Animals were allowed to settle by gravity, buffer was aspirated, and animals were washed with 1x PBST twice. Worms were then resuspended in 60% isopropanol and incubated 15 min with gentle rocking at room temperature to dehydrate. Worms were allowed to settle by gravity and remaining isopropanol was removed. 1 ml of 60% ORO was added and the worms were incubated overnight with gentle rocking. After the overnight incubation, ORO was removed and worms were resuspended in 100 μl of 1x PBST. 1 μl 1x DAPI was then added to the 20 μl PBST, as an internal control for permeabilization, and flicked to mix. Worms were then mounted on slides on an agarose pad and visualized with Zeiss LSM 700 confocal microscope using either a Plan-Apochromat 10x/0.45 or Plan-Apochromat 20x/0.8 objective lens, as described below.

## Carminic acid staining

Carmine dyes such as carminic acid are used to complex with glycogen and short glucose polymers to produce a fluorescent signal responsive to the levels of glucose polymer. Thus, it is a useful tool to analyze glycogen storage [22]. Carminic acid was introduced to worms through feeding as previously described [22]. Briefly, OP50 was inoculated into LB broth with 1 mg/ml carminic acid and grown overnight at room temperature in a light protected vial. OP50 with carminic acid was then used to seed standard NGM plates. Animals were plated on carminic acid containing plates following initial mid-L4 harvest (controls) or post-ARD at day 5, day 10, and day 30 and were visualized with Zeiss LSM 700 confocal microscope using a Plan-Apochromat 20x/0.8 objective lens, as detailed below.

## Microscopy & image analysis

Microscope images were acquired on a Zeiss LSM 700 confocal microscope using either Plan-Apochromat 10x/0.45 or a Plan-Apochromat 20x/0.8 objective lens. In all cases: photomultiplier tube (PMT) offset was adjusted so that zero photons corresponded to a pixel value of zero; and PMT gain was adjusted to fully utilize the available 8-bit dynamic range while avoiding saturating the images. The same settings were used across all images being compared.

When acquiring the (non-fluorescence) ORO images, sequential scans were performed with the 488nm, 555nm, and 639nm lasers while using the transmitted light detector to produce an RGB image like what one would acquire using a color camera under widefield illumination from white light. Differential Interference Contrast (DIC) optics were also in place while imaging to provide contrast sufficient to distinguish sample anatomy at the same time as recording ORO. Either the 10x/0.45 or the 20x/.8 objective lens was used for ORO imaging—there was no difference in quantitation because the numerical aperture of the condenser was limited to <0.45, and kept constant across experiments.

When acquiring fluorescence images of carminic acid, a 555nm laser was used for excitation. The emission was passed through a laser blocking filter and a wide open pinhole to produce a non-confocal image. The emission was further filtered through a SP640 (640 nm short pass) emission filter, to produce a widefield fluorescence image appropriate for quantitating

total signal from the sample volume. Simultaneously with the fluorescence channel, a DIC image was also acquired. Only images acquired with the 20x/0.8 lens were used for quantitation.

For quantitative image analysis, Fiji software [34] was used, employing in-house macros customized to streamline the processing for the user. For both ORO and carminic acid, regions of interest (ROIs) were manually traced very roughly around the edges of each worm, further refined by automatic segmentation, then the average intensity per pixel was calculated. This mean pixel value was divided by the diameter of the worm, measured manually by the user, to provide a relative value for quantitating mean carminic acid fluorescence per unit volume. For the ORO (RGB) images, a more in-depth analysis was performed, converting the intensities to relative absorbance (pseudo-fluorescence) at each pixel location, then combining with automated physical measurements of the worm from the ROI, to calculate an average ORO density per unit volume. Quantitative comparisons could then be made (S1 Methods). For presentation, all images had their intensities re-scaled to maximize the dynamic range and visual appeal of the images. Images of ORO and fluorescence that were being compared were rescaled by the same amount in those channels.

## Results

### Diverse nutrient sensors are required for ARD entry

In the initial characterization of ARD, only one allele impacting ARD entry was identified:the *nhr-49(nr2041)* mutant allele of NHR-49, a protein involved in fatty acid metabolism [13]. Subsequent studies have identified additional pathways which regulate ARD in a variety of ways, and one has called into question the impact of *nhr-49* on ARD entry [35, 36]. We sought to identify additional regulators of ARD dynamics, focusing on the three distinct phases of ARD: initiation (proportion of animals entering ARD vs bagging or L4 fate), maintenance (germline shrinkage and retained embryos), and recovery (post-ARD lifespan, germline regrowth, and brood size) [13] (Fig 1A). Using a candidate gene approach, we focused on key nutrient-responsive pathways. The pathways profiled included AMP/energy signaling (AAK-2/AMPK), insulin-like signaling (AGE-1/PI(3)K, DAF-16/FoxO), TOR (RSKS-1/S6K), stress signaling (SKN-1/Nrf), carbohydrate metabolism/hexosamine signaling (OGT-1/OGT and OGA-1/OGA), and sirtuins (SIR-2.1/SIRT1). The alleles analyzed were chosen based on their viability under the experimental conditions and that they have been previously characterized for their roles in nutrient signaling and longevity.

ARD initiation was clearly discernible by 48 to 72 hours after mid-L4 worms were placed on ARD plates and could be distinguished from the bagging or L4 fates by the morphology of the worms, particularly the presence of one to two embryos retained in the uterus of adult worms [13] (S1 Fig). We found that *age-1(hx546)*, *daf-16(mu86)*, *rsks-1(ok1255)*, and *skn-1 (zj15)* did not significantly alter ARD initiation, suggesting that insulin-like signaling, the TOR pathway, and stress response are not major contributors to entry (Fig 1B), in agreement with another recent report [35]. In contrast, *aak-2(ok524)*, *ogt-1(ok430)*, *ogt-1(ok1474)*, *sir-2.1 (ok434)*, and our positive control *nhr-49(nr2041)* all contribute to the initial sensing and response to starvation during ARD entry. These mutants displayed a significant reduction in the ability of nematodes to successfully initiate ARD (Fig 1B, S2 Fig).

We were intrigued to find that *ogt-1* mutants exhibited severe ARD entry defects similar to *nhr-49(nr2041)* (Fig 1B, S2 Fig). We verified this finding by generating and analyzing targeted full gene deletions, *ogt-1(jah01)* and *oga-1(av82)*. These new CRISPR/Cas9 alleles were confirmed to have similar phenotypes as the previously characterized *ogt-1* and *oga-1* alleles, including lifespan and brood size under normal laboratory conditions (S1 Table). As with the

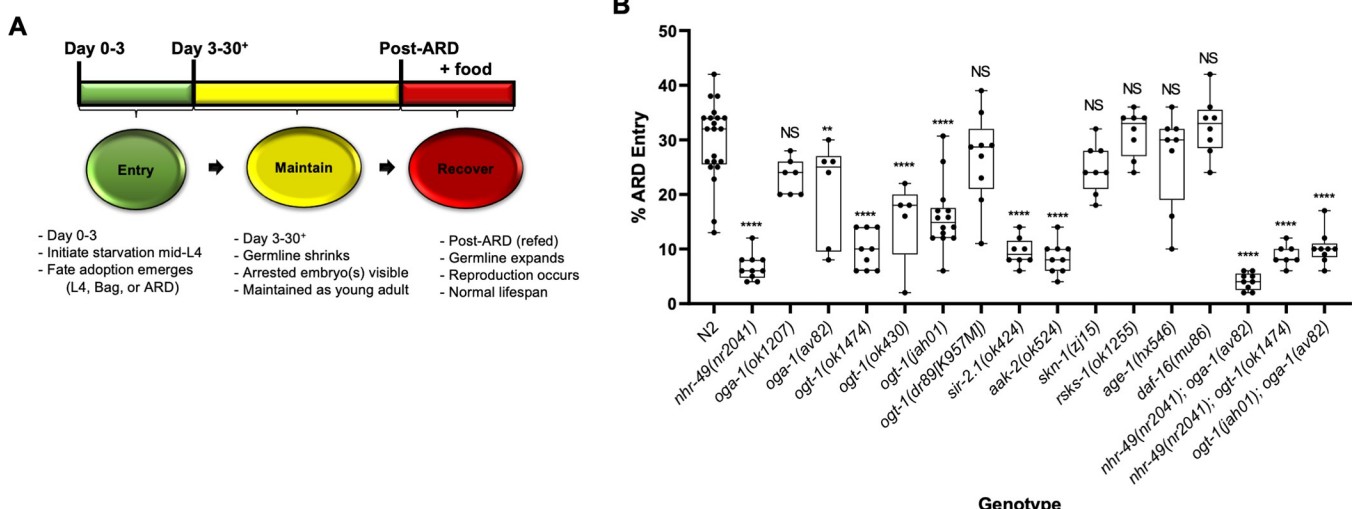

**Fig 1. ARD stages and entry defects in nutrient sensing mutants.** (A) ARD stages including initiation, remodeling/maintenance, and recovery. Initiation occurs when worms are removed from a food source during mid-L4. With wild type worms, approximately 1/3 enter ARD, 1/3 have hatched offspring inside (bagging), and 1/3 maintain an L4 state during the first 72 hours of starvation. Once ARD is initiated, worms enter the maintenance phase wherein the germline shrinks and the characteristic retained embryo(s) are observed. Recovery occurs after starved worms are re-fed. During this stage the germline expands and a normal lifespan commences (adapted from [13]). (B) The percentage of worms entering ARD as opposed to bagging or L4 fate were quantified for the respective mutants. For single mutants ARD initiation was significantly defective in the following strains, as compared to N2: *nhr-49(nr2041)*, *aak-2 (ok524)*, *oga-1(av82)*, *ogt-1(jah01)*, *ogt-1(ok430)*, *ogt-1(ok1474)*, and *sir-2.1(ok424)*. The double mutants *nhr-49(nr2041);oga-1(av82)*, *nhr-49(2041);ogt-1 (ok1474)* and *ogt-1(jah01);oga-1(av82)* also showed low ARD entry. A minimum of 300 worms from 3 different experiments were counted and statistics were performed in Graph Pad Prism using an Ordinary one-way ANOVA. *P*-value **** = <0.0001, ** = 0.001, NS = not significant. Box and wisker plots show median, quartiles, and range.

other two *ogt-1* alleles, we observed a significant ARD entry defect with *ogt-1(jah01)* (Fig 1B). Because of this, we used the *ogt-1* alleles interchangeably. Both the *oga-1(ok1207)* and newly-generated *oga-1(av82)* alleles show a decrease in mean ARD entry, but this effect was milder than for *ogt-1* and was only significant with the CRISPR allele (Fig 1B, S2 Fig). For this reason, we continued with the full gene deletion *oga-1(av82)* for further analysis. The milder and allele-specific phenotypes of the *oga-1* mutants suggested that removal of *O*-GlcNAc does not have as strong of an influence on ARD as addition of *O*-GlcNAc. Considering this and recent findings demonstrating non-catalytic roles of OGT-1 in *C. elegans* [37, 38], we tested ARD entry of worms with the catalytic-dead mutation *ogt-1(dr89[K957M])*. Interestingly, we found that this point mutant line entered ARD at similar rates to N2, demonstrating this starvation response phenotype does not require the ability of OGT-1 to transfer *O*-GlcNAc, suggesting it instead depends upon a non-catalytic role of the protein (Fig 1B).

Associated with ARD initiation, we noted significant differences between the mutant strains in the distribution of animals adopting either the L4 or bagging fate. Both L4 and bagging had a higher degree of variability than ARD entry, in general, suggesting that the ARD entry may be more tightly regulated than the other two fates (S2 Fig). Interestingly, some of the mutant alleles showing reduced ARD entry (*nhr-49(nr2041)*, *ogt-1(ok1474)*, and *oga-1(ok1207)*) did so while producing sizable numbers of animals adopting both the L4 fate and bagging. The other strong entry defective alleles, *aak-2(ok524)* and *sir-2.1(ok434)*, produced predominantly L4 arrested larvae with few bagging or in ARD (S2 Fig), suggesting there may be alternative mechanisms at play in the regulation of entry between mutants and, therefore, distinct pathways.

Based on the similarity of phenotypes for the single mutants of *nhr-49* and *ogt-1*, in addition to their known overlapping roles in metabolism, we next chose to explore the genetic interaction between these mutants by analyzing the ARD initiation phenotypes in double mutants.

Three double mutant lines were generated to test the interactions between *nhr-49*, *ogt-1*, and *oga-1*. *nhr-49(nr2041);oga-1(av82)*, *nhr-49(nr2041);ogt-1(ok1474)*, and *ogt-1(jah01);oga-1 (av82)* each showed severe ARD entry defects similar to *nhr-49(nr2041)* single mutants (Fig 1B). Because the ARD entry rate is so close to zero for *nhr-49* and *ogt-1* mutants, it was difficult to discern between an additive effect of parallel pathways or the effect of the more severe allele in a single pathway. Therefore, our epistasis experiments were unable to determine whether *nhr-49*, *oga-1*, and *ogt-1* function in the same pathway to regulate ARD entry or each have independent roles.

## Distinct nutrient sensors are required for ARD recovery

Successful maintenance of ARD can be determined by examining germline shrinkage and embryo preservation that occurs after entry into ARD. Characteristics of the recovery stage of ARD is marked by dynamic germline regrowth, production of progeny (brood size), and post-ARD lifespan [13]. Thus, to further define which nutrient sensors regulate each of these stages of ARD, we next looked at the dynamics of the germline, brood size, and lifespan.

Despite significant changes in ARD entry, we found that all mutant strains exhibited robust germline shrinkage once ARD was established, analogous to wild-type worms and the retained embryos were observable throughout the starvation period (Fig 2, left panel]. Further, when worms were re-fed for two days following 30 days of starvation (post-ARD), a robust regrowth of the germline was observed in all strains (Fig 2, right panel). Thus, in this set of nutrient sensors, dynamic germline shrinkage and regrowth is a general feature of ARD once it is established, as is the maintenance of the retained embryos [13, 36]. These results suggest that the maintenance of ARD and germline shrinkage may be independent of nutrient sensing and support the hypothesis that the distinct phases of ARD may be genetically separable [13].

As noted previously, post-ARD animals retain their ability to reproduce, albeit to a significantly diminished degree. However, this was found to be largely dependent on the amount of sperm retained in the hermaphrodites, with selfing producing significantly less offspring than mating post-ARD [13]. In our analysis of brood size following ARD, we found that there was a high level of variability in the number of progeny, even between replicates of the same strains within the same experiment (S1 Table). Surprisingly, reduced brood size did not correlate with reduced sperm counts (S3 Fig) nor with the germline dynamics we observed (Fig 2). This suggests that the regulation of ARD recovery as measured by brood size is complex and may not correlate to sensing and responding to nutrient changes but may involve other processes. We therefore chose to focus on post-ARD lifespan as a measure of ARD recovery.

Post-ARD lifespan is a means to quantify the survivability of ARD, or the ability to recover and thrive following re-feeding after time spent in ARD [13]. Thus, we analyzed the lifespan of worms recovered after 30 days of ARD using standard lifespan assays. We found that *aak-2 (ok524)*, *daf-16(mu86)*, *rsks-1(ok1255)*, and *skn-1(zj15)* mutants had significantly reduced post-ARD lifespans (Fig 3). A concurrent study also found these genes reduced post-ARD lifespan, with the exception of *skn-1* [35]. Based on tour analysis, we suggest that the TOR pathway most impacts recovery.

## Mitochondrial β-oxidation but not fatty acid desaturation is required for ARD entry

Due to the strong phenotypes we observed for ARD entry, the role for *O*-GlcNAc signaling, and the similarity in phenotypes for *nhr-49* and *ogt-1* mutants, we next explored this entry pathway in more detail. We chose to focus on *ogt-1* and not *oga-1* for the rest of the experiments due to the variability of *oga-1* alleles and the weaker entry phenotype.

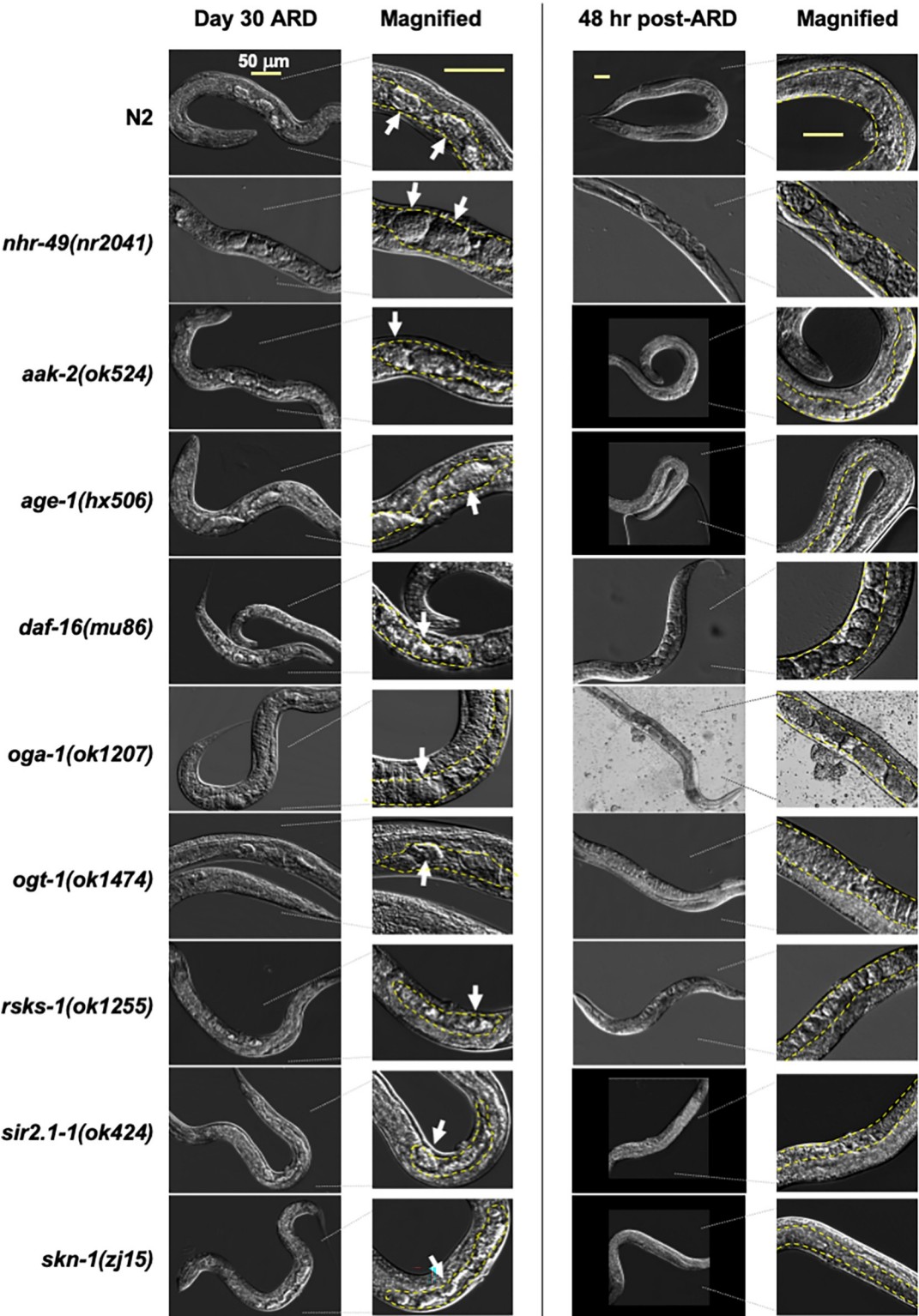

**Fig 2. Germline shrinkage during ARD maintenance and regrowth during recovery.** Representative images (with magnified panels for detail] showing the germline (outlined in yellow] of worms starved (left panel] at day 30 of ARD, with uterine embryos noted with white arrows. The right panel shows worms that have been re-feed for 48 hours after 30 days of ARD, with white dashed lines to indicate the germline, where healthy embryos and/or oocytes can be seen. All strains showed robust shrinkage of the germline during starvation and regrowth after feeding. The 50 μm scale bar shown in the top row applies to all images in each column.

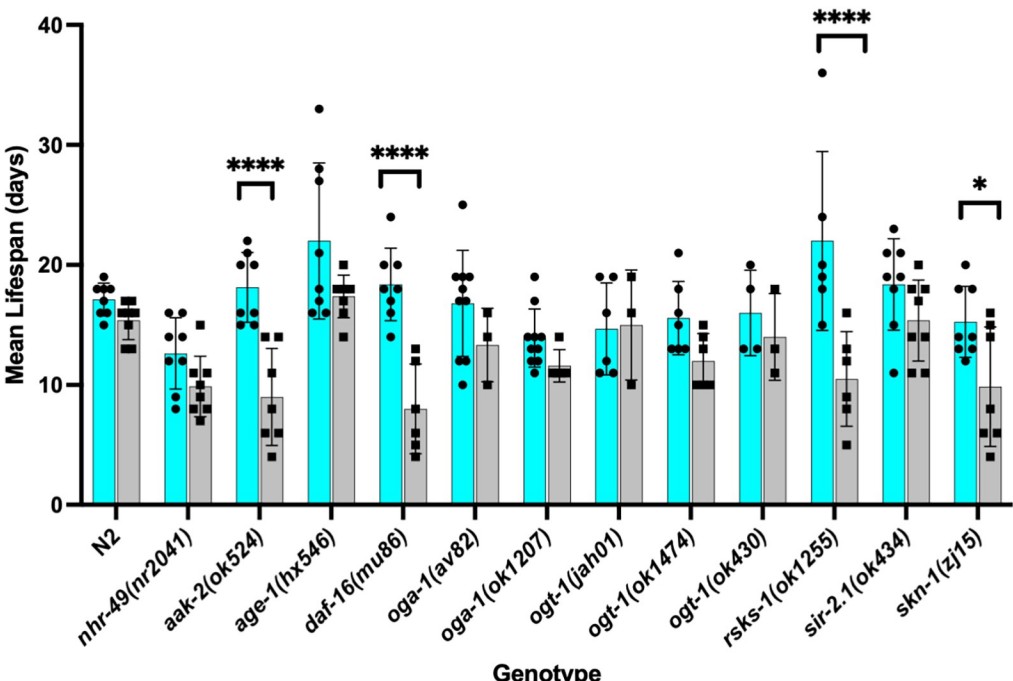

**Fig 3. Lifespan of mutant strains in recovery after 30 days of ARD and summary model.** Lifespan of wild-type and mutant strains in control husbandry conditions (blue bars) compared to those recovered after 30 days of ARD (gray bars). Lifespan after recovery from ARD was significantly defective in *aak-2 (ok524)*, *daf-16(mu86)*, and *rsks-1(1255)*. The mutant *skn-1(zj15)* also had a significantly reduced lifespan post-ARD but not as strongly as the others. A minimum of 8 worms from independent experiments were analyzed for lifespan and statistics were performed in Graph Pad Prism using one-way ANOVA. *P*-value **** = <0.0001, * = 0.0448. Error bars represent standard deviation. Overall, we found that whereas *nhr-49*, *oga-1*, *ogt-1*, and *sir-2.1* play a role in ARD initiation, *daf-16*, *rsks-1*, and *skn-1* function primarily in recovery from ARD, with *aak-2* playing a role in the regulation of both states. Conversely, none of the alleles that we explored significantly impacted maintenance of the ARD.

NHR-49 transcriptionally regulates both fatty acid desaturation via *fat-7*, and mitochondrial β-oxidation via *acs-2* [39] (Fig 4A). To determine if these NHR-49-dependent pathways is relevant for ARD entry we analyzed *acs-2(ok2457)* (mitochondrial beta-oxidation) and *fat-7 (wa36)* (fatty acid desaturation) mutants for ARD phenotypes. Whereas the *acs-2* mutant and *acs-2;ogt-1* double mutant had a significant defect in ARD entry, mutant *fat-7* entry was comparable to wild-type. Similar to the *nhr-49*, *oga-1*, and *ogt-1* single mutants, we did not observe a significant difference in lifespan of the *fat-7* or *acs-2* mutants post-ARD (Fig 4C). This data diverges slightly from a report which showed *acs-2* and *fat-7* mutants had reduced post-ARD lifespan [35]. These effects were relatively minor, and *fat-7* was not tested alone but in combination with a *fat-6* mutation [35], which may explain why our findings differ. Our findings further implicate lipid metabolism and suggest that *acs-2* regulates ARD entry to a greater extent than recovery.

## Neutral lipid changes associated with ARD entry

Changes in lipid metabolism play a prominent role in multiple pro-longevity pathways [40]. NHR-49 and the *O*-GlcNAc cycling enzymes have well-established roles in the regulation of

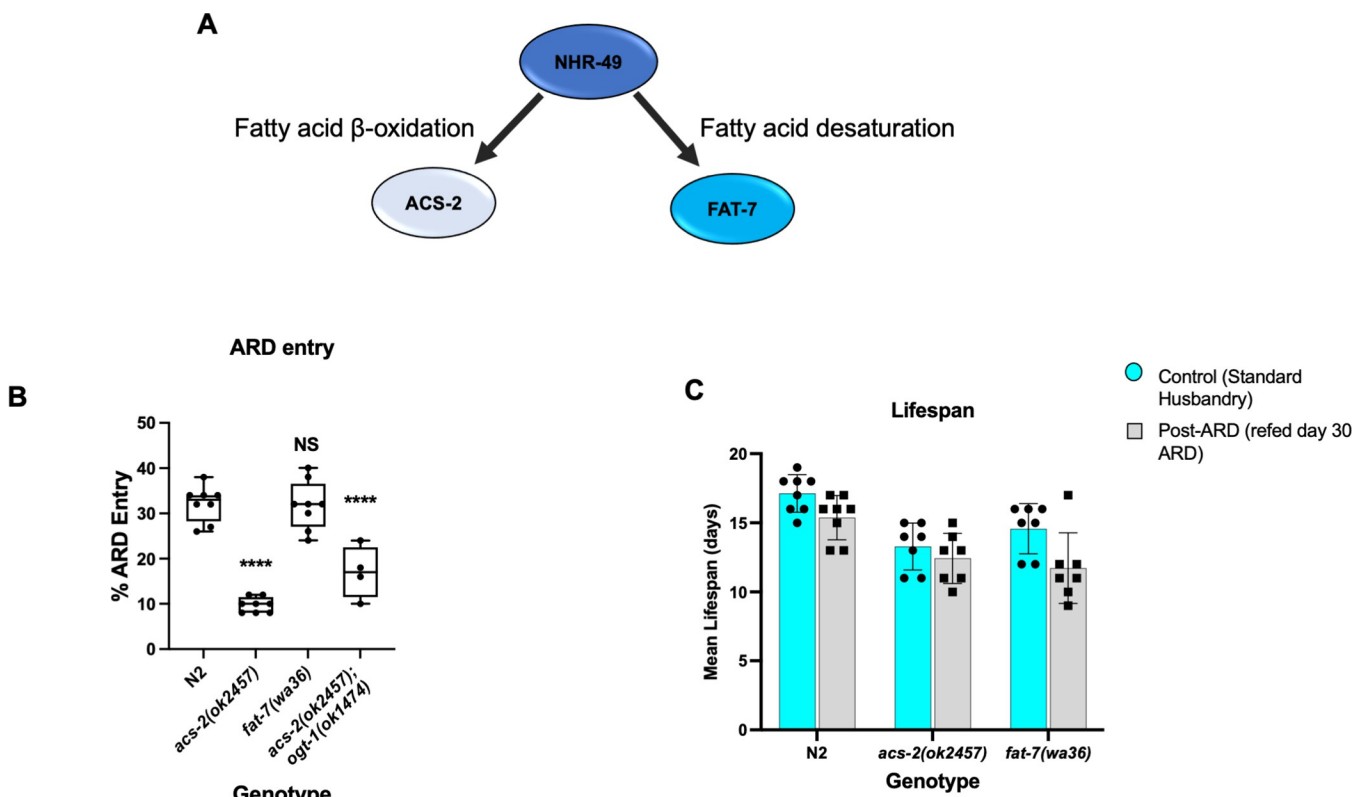

**Fig 4. Fatty acid beta-oxidation, but not desaturation, plays a role in ARD entry.** (A) NHR-49 regulates both fatty acid beta-oxidation (ACS-2) and desaturation (FAT-7) via genetically separable pathways that are known to generate distinct phenotypic outputs. (B) ARD entry was significantly impaired with *acs-2(ok2457)* but not with the *fat-7(wa86)* mutant. The double mutant *acs-1(ok2457); ogt-1(ok1474)* was also significantly impaired for ARD entry. *P*-value **** = <0.0001 in comparison with N2 as determined using one-way ANOVA, NS = not significant. (C) Lifespan under standard conditions and with refeeding at day 30 of ARD was not significantly altered in these mutants as determined by one-way ANOVA. Error bars represent standard deviation.

fatty acid metabolism [39, 41, 42]. To determine the effects of ARD-regulating pathways on lipid storage, we used the lysochrome dye Oil Red O (ORO), which stains triglycerides (TAGs) [33]. We examined the various strains for ORO staining in control (fed conditions) and after 30 days in ARD (prolonged exposure). Fed *ogt-1(ok1474)* worms showed significantly lower ORO staining compared to N2s (Fig 5A and 5B), as has been reported previously with *ogt-1* mutants [22]. ORO staining is present throughout the body of worms in fed conditions, but after 30 days in ARD, staining is greatly diminished throughout the body with the exception of the retained embryo (Fig 5A and 5B). Between the fed and ARD conditions, *ogt-1(ok1474)* animals did not show a significant decrease in quantified ORO staining, likely due to the already-low lipid stores in this strain. Additionally, the quantification was normalized by volume (see S1 Methods), which may underestimate the effect of ARD on total triglycerides per animal, as worms in ARD are significantly smaller than fed worms.

Notably, we observed strong depletion of lipid stores as early as 24 hours of starvation (S4A Fig), suggesting it is an early response. This is supported by another study which also noted decreased ORO signal in ARD worms at 48 hours of starvation [35]. As a control, we also monitored cuticle penetrance of a small molecule dye using DAPI which revealed comparable penetrance between strains (S4B Fig). We further observed that in other strains analyzed, those without major effects on ARD entry (*age-1(hx546)*, *fat-7(wa36)*, and *skn-1(zj15)*) showed higher levels of ORO staining throughout the animal in ARD than wild-type worms in ARD

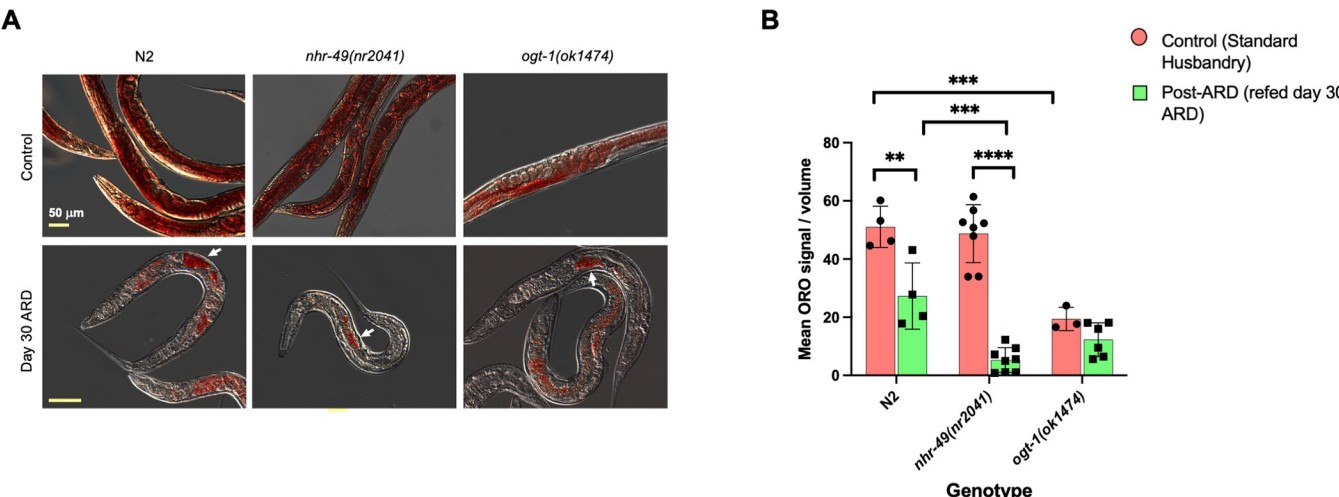

**Fig 5. Oil Red O staining reveals redistribution of lipid stores in ARD.** (A) Representative images showing the distribution of Oil Red O in control (top) and starved animals after day 30 of ARD (bottom) in the *ogt-1*-dependent pathway. Retention of TAGs appeared to be strongest in the retained embryos in all mutant strains (white arrows). Scale bar indicates 50 μm and is consistent across each row. (B) Fiji Image analysis-based quantification of ORO staining (see S1 Methods) of wildtype and the mutant strains of the *ogt-1*-dependent pathway in normal husbandry (control) and in day 30 of ARD. Using two-way ANOVA analysis, both variables (genotype and feeding condition) were found to have significant effects, and these variables interact significantly. For each strain, mean ORO density decreased after 30 days of ARD: most dramatically for *nhr-49(nr2041)* and not significantly for *ogt-1(ok1474)*. In normal husbandry conditions, *ogt-1(1474)* showed significantly less ORO signal than did N2 worms. *P*-value **** = <0.0001, *** = <0.001, ** = <0.01. Error bars represent standard deviation.

(Fig 5, S5 Fig). Strains with significant ARD entry defects (*acs-2(ok2457)*, *sir-2.1(ok424)*, and *oga-1(tm3642)*) showed reduced ORO staining when in ARD, with characteristic retention of lipids in the one or two uterine embryos (S5 Fig). This correlation suggests lipid stores play an important role in the initiation of this form of diapause.

Based on the interplay between carbohydrate and fatty acid metabolism, as well as the role of *O*-GlcNAc in carbohydrate metabolism, we next looked at glycogen storage using carminic acid staining. Under control husbandry conditions, our analysis suggested a change in the pattern of glycogen storage in *nhr-49(nr2041)* and *ogt-1(ok1474)* worms, despite fluorescent signal having no significant difference between strains (S6 Fig). After 30 days of ARD, these stains all showed higher carminic acid signal normalized by volume. This may be partially attributable to the reduction in size associated with ARD, and suggests glycogen stores are relatively unaffected or increased by ARD. Interestingly, *nhr-49(nr2041)* worms in ARD showed dramaticly higher carminic acid staining (a 3-fold increase compared to N2) than did non-ARD worms (S6 Fig). With the findings of lipid reduction in ARD, this suggests glycogen stores may be maintained or increased at the expense of lipid stores while in ARD. These results suggest ARD has effects on glycogen storage, and glycogen storage does not correlate with the ARD entry defects we see in *nhr-49* and *ogt-1* mutant strains.

## Discussion

Adding to the growing literature on adult reproductive diapause, we have demonstrated roles for multiple nutrient-sensing pathways which regulate this process. Our results showed that *aak-2*, *ogt-1*, *oga-1* and *sir-2.1* regulate ARD entry, thus implicating fat metabolism, the hexosamine/*O*-GlcNAc cycling pathway, and energy status in the initial sensing and dissemination of the ARD entry signal. Though all *ogt-1* alleles tested exhibited strong ARD entry defects, the two *oga-1* alleles were much milder, with only the CRISPR allele being significantly lower than wild-type. Thus we investigated whether OGT-1 catalysis was required for the entry defect.

Interestingly, we found that ARD entry did not require the ability of OGT-1 to transfer *O*-GlcNAc. Interactions between OGT and these other nutrient-sensing pathways have been documented in other systems, for example, human homologs of AAK-2 (AMPK [21]), SIR-2.1 (SIRT1 [43]), and NHR-49 (PPAR [44]) are reported to be *O*-GlcNAc modified. Further investigation will be necessary to determine if the pathways impacting ARD entry are interconnected, and the mechanisms by which they impact each other. While our data showing that OGT-1 catalysis was dispensable for normal ARD entry it is possible that physical interactions between OGT and these proteins are essential for proper entry.

ARD maintenance, which was characterized by the shrinkage of the germline and retention of embryo(s) over time during ARD, was not significantly altered in any of the strains analyzed (Fig 2). Previous research found that the apoptosis factor CED-3 was required for ARD germline shrinkage [13], though this role has been questioned by later studies. Carranza-Garcia and Navarro found that while CED-3 was important in maintaining oocyte quality, it was not required for germline shrinkage, positing instead that an ovulation-based mechanism is involved [36]. Our results herein suggest that maintenance of ARD is independent of nutrient sensing and instead relies on other processes which establish and maintain the morphological and molecular characteristics of this state once the organism has committed to enter ARD.

ARD recovery, as assessed by post-ARD lifespan, was disrupted in *aak-2*, *daf-16*, *rsks-1*, and, to a lesser extent, *skn-1* mutants (Fig 3). While *daf-16* is most often associated with insulin-like signaling, it is also known to regulate and be regulated by the AMPK and TOR pathways [45]. Based on the phenotypic similarity between *rsks-1* (TOR) and *daf-16*, with our finding that *age-1*, which acts upstream of *daf-16* in insulin signaling, doesn't impact ARD, we speculate that the TOR pathway is primarily responsible for recovery from ARD, particularly as loss of *age-1*, did not have a significant impact on any of the ARD phases (Figs 1B, 2 and 3). This fits well with the findings of a recent article which demonstrated that both hypo- and hyper-activation of TOR signalling reduced post-ARD lifespan [35], suggesting fine-tuning of this pathway is important for proper ARD exit. Further, *aak-2*/AMPK can also influence *daf-16*, and we find that *aak-2* had significant impacts on both entry and exit, a distinct phenotype from *daf-16* alone (Figs 1B, 3). It is interesting to see both TOR and AMPK regulate ARD, as these two nutrient-sensing pathways have considerable interplay [46, 47], the balance of which may be important in the commitment to enter ARD and in later returning to a normal reproductive adulthood state. Gerisch et al. also identified *aak-2* as an essential player in ARD, that is both required for extended lifespan after ARD, and being a target of their newly identified ARD "master regulator" *hlh-30* [35]. Four genes we identified as affecting ARD entry, *aak-2*, *acs-2*, *nhr-49*, and *oga-1*, are predicted *hlh-30* targets based on ChIP data [48], suggesting this transcription factor may act upstream of these pathways and influence ARD entry in part through transcriptional regulation of these genes.

Overall, our findings support the previous assertion that ARD phases may be largely genetically separable [13], with a general role for sensing changes in energy status both to enter and exit ARD (Fig 6). This complex constellation of regulators and distinct stages of ARD offers a promising model for future exploration of the contribution of these pathways in the regulation of longevity and may shed light on some of their dynamic roles under various pro-longevity regimens.

The *O*-GlcNAc transferase is the sole enzyme responsible for the intracellular addition of *O*-GlcNAc, a highly dynamic nutrient-sensing post-translational modification (PTM) that allows for a rapid and reversible response to metabolic fluctuations [24, 42].The importance of *O*-GlcNAc in regulating developmental diapause in *C. elegans* has been explored in previous studies, which have shown *ogt-1* and *oga-1* regulate the dauer state [22, 23]. Here, our results demonstrate that *ogt-1* has a distinct role in enabling entry into ARD, another diapause state

## ARD ENTRY AND RECOVERY PATHWAYS

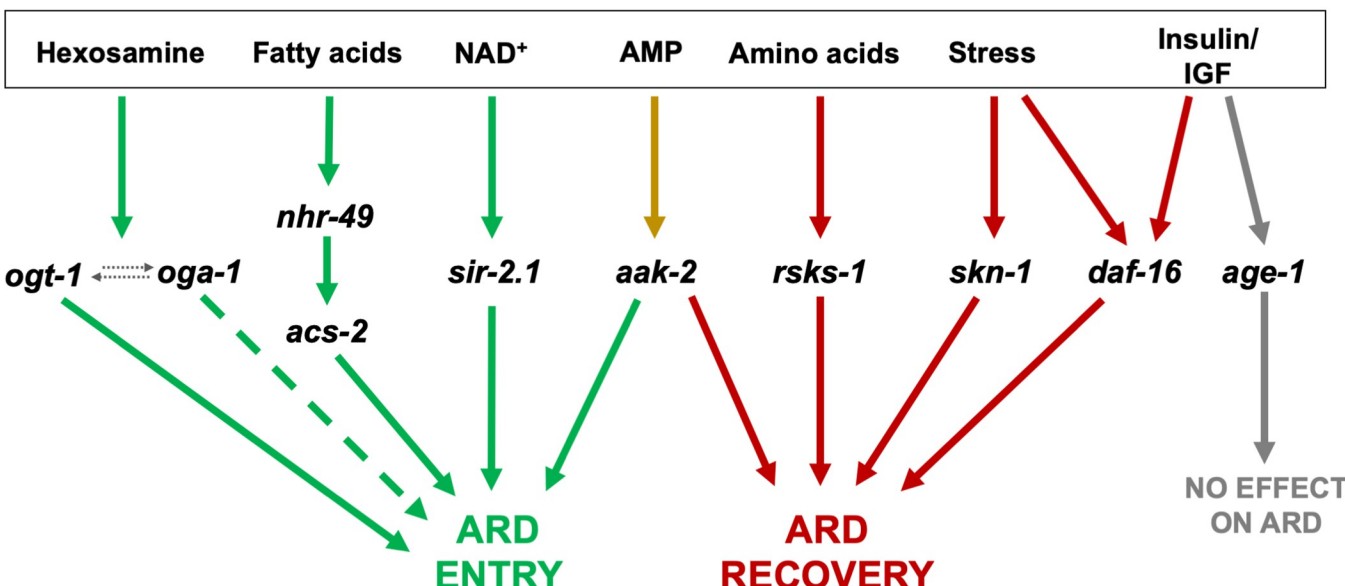

**Fig 6. Model of genetic modifiers of ARD phases.** Specific nutrient-sensing pathways contribute to ARD entry and exit. The initial sensing and dissemination of the signal to enter ARD (green pathways) includes genes encoding proteins involved in *O*-GlcNAc cycling (*ogt-1*, *oga-1*, with dotted arrows showing their biochemical interaction), genes that regulate fatty acid metabolism (*nhr-49*, *acs-2*) and sirtuins (*sir-2.1*). The dashed green line indicates an allele-specific effect on ARD entry. Energy sensing, mediated by the product of *aak-2*, uniquely acts in both ARD initiation and recovery (yellow pathway). The two members of the insulin/IGF signaling pathway tested in this study showed differing results with *age-1* not affecting ARD (grey pathway), while the downstream *daf-16* affected recovery (red pathway). ARD recovery additionally relies on the TOR pathway (*rsks-1*), which may also interact with *daf-16*. Stress signaling influences ARD recovery as well, with *skn-1* playing a minor role, and another possible pathway interacting with *daf-16*.

later in development which is controlled by different genetic and environmental factors [13, 35]. Further, loss of either *ogt-1* or *oga-1* each reduced ARD initiation (Fig 1B), unlike the regulation of dauer entry, in which the two genes have opposing effects. This suggests that while the *O*-GlcNAc cycling enzymes regulate both diapause states, they do so through different pathways. Indeed, our results indicated that the role of OGT-1 in ARD entry does not require its *O*-GlcNAc transferase activity (Fig 1B). Though dauer entry has not yet been tested to see if OGT-1 catalysis is required, the opposite effects of *ogt-1* and *oga-1* deletions suggest that process may indeed be mediated by *O*-GlcNAcylation rather than a non-catalytic role OGT-1, which future studies should address. It is also possible that OGA-1, may also have additional functions independent of *O*-GlcNAc removal which may be involved in ARD. For example, it has been suggested that the *lin-4* microRNA controls the levels of both *sbp-1* and *oga-1* to regulate fat metabolism, longevity, ROS production, and locomotion [49].

Both OGT-1 and OGA-1 are known to physically interact with several proteins and can form complexes that alter the localization and function of other proteins [50–54]. Thus, OGA-1 and OGT-1 both may have other non-*O*-GlcNAc cycling functions, such as physically interacting with proteins in a way that alters their activity. Our data suggests that the impact of OGT-1 on ARD entry is not due to its catalytic activity as a glycosyltransferase. OGT-1 could be involved in sensing low nutrient levels to trigger ARD, or otherwise involved in establishing the diapause state. A nutrient-sensing function may still rely upon cellular UDP-GlcNAc concentration, as these levels alter OGT's targets [55] and therefore may alter its protein-protein interactions. Future studies are necessary to further tease apart OGT-1's catalytic and non-catalytic functions.

The ARD entry defects of *ogt-1* and *nhr-49* strains correlate with changes in fat stores (Fig 5A) but not carbohydrate stores (S6 Fig), which could suggest these genes are acting in a common or interrelated pathways. Thus, our genetic and staining results support the hypothesis that modulation of fat stores is a key factor in ARD, as others have proposed [13, 35].

NHR-49 regulates fatty acid metabolism by triggering fatty acid β-oxidation and/or desaturation [39]. Several desaturases and unsaturated fatty acids are essential under different pro-longevity regimens, with multiple lines of evidence supporting an important role for lipid metabolism in pro-longevity pathways [9, 56, 57]. NHR-49 has a broad role in longevity, such that worms with impaired NHR-49 have decreased longevity under normal, fed conditions. This reduced lifespan under standard conditions is due to decreased fatty acid desaturation [39]. On the other hand, both desaturation and β-oxidation of fatty acids mediated by NHR-49 is important for the increased longevity related to loss of the germline [58]. Our results demonstrate that ARD entry requires the β-oxidation pathway (*acs-2*) downstream of *nhr-49*. A recent study reported *nhr-49* was not required for ARD [35], contrary to both the initial characterization of ARD [13] and the data we present here. The Gerisch et al. study induced ARD by starving worms at the mid-L3 stage [35] rather than mid-L4 which we used in our experiments, following the original methodology reported by Angelo and Van Gilst [13]. Thus, these contradictory results between studies may reflect differences in how commitment to ARD entry occurs in different developmental stages, with *nhr-49* playing an important role when L4, but not L3 larvae are exposed to starvation conditions. Importantly, our study agrees with Gerisch et al. on the effects of many genes with ARD regulatory functions such as *rsks-1*, *aak-2*, *acs-2*, and *daf-16*.

The regulation of fatty acid metabolism by NHR-49 has important implications for longevity, the mechanism of which may be context-dependent and dynamic. OGT-1 is uniquely situated at the node of multiple longevity pathways [24, 42], and the regulation of ARD entry described here adds to the complex interplay of nutrient signaling with development and longevity. Our analysis of the genes encoding the *O*-GlcNAc enzymes and NHR-49 suggests a complex dynamic is at play. One interpretation of the findings is that OGT-1 may act as a metabolic rheostat which regulates the switch between fatty acid oxidation and desaturation, depending on environmental conditions. Previous work [39, 41] has indicated that during starvation, mitochondrial β-oxidation (*acs-2* and other genes) increases fat consumption at the expense of *fat-7*-dependent fatty acid desaturation. We have also previously examined *fat-7* and *acs-2* expression in the context of the *ogt-1* and *oga-1* mutants and their role in innate immunity and starvation where deregulation of both of these genes was observed. In the same paper, we reported that both *ogt-1* and *oga-1* mutant L4s had no change in the transcription of *nhr-49* [29]. These gene expression findings fit our genetic evidence as lipid stores are depleted in strains defective in ARD entry and *acs-2* seems to be critical for this utilization of stored fats under these conditions, whereas the *fat-7* mutants did not alter ARD initiation (Fig 4B).

## Conclusion

In this study, we have placed the *O*-GlcNAc cycling enzymes in the context of other well established pro-longevity-associated nutrient sensors in modulating adult reproductive diapause. We have identified several regulators of ARD initiation including *sir-2.1*, *aak-2*, *acs-2*, *ogt-1* and *oga-1*, in addition to the previously described role for *nhr-49*. Further, we have shown that *skn-1*, *rsks-1*, and *daf-16* are important for ARD recovery. Defining key metabolites that are altered will provide additional insight into the central components regulated by the OGT-1 and how these are altered under specific environmental conditions. Future exploration examining how *O*-GlcNAc cycling enzymes interact with nuclear hormone receptors such as NHR-

49 may provide new insight into the role of OGT and OGA as metabolic rheostats. The findings described herein indicate that ARD entry and recovery are largely genetically separable and that OGT-1acts with other nutrient sensors in ARD initiation. These results have important implications for understanding the dynamic activity of nutrients sensors during dietary restriction and for defining the role of these sensors in regulation of mammalian lipid mobilization and metabolism.

## Supporting information

**S1 Fig. Summary of ARD protocol.** As described in Materials and Methods, worms were bleached to obtain synchronized L1 larvae. The L1s were then plated at a density of 10,000 per plate, harvested at mid-L4 stage, and grown on plates without food for 48–72 hours. At this stage worms were counted to determine the percentage of animals in arrested L4, bagged adults, or ARD (arrows indicate retained embryos) as shown in the representative photos. (TIF)

**S2 Fig. Fates adopted by each strain during initiation of ARD.** Percent of worms from each respective genotype in L4 (blue bar), bag (purple bar) or ARD (grey bar) are represented. The L4 and bagging fates tended to have a higher degree of variability than ARD entry, as evidenced by a higher standard deviations. Select strains had a marked increase in the percentage of worms in L4 including *aak-2(ok524)*, *age-1(hx546)*, *daf-16(mu86)*, *sir-2.1(ok434)*, and *skn-1 (zj15)*. Other mutant strains, *nhr-49(nr2041)*, *oga-1(ok1207)*, and *ogt-1(jah01)*, were more similar to wildtype in terms of having a more even distribution between L4 and bagging even though this set was defective for ARD entry. (TIF)

**S3 Fig. DAPI staining of spermatheca during ARD reveal changes in number of sperm nuclei during ARD.** (A) DAPI staining of worms after 15 days in ARD reveals fewer sperm in the spermatheca of *ogt-1(1474)*, and *nhr-49(nr2041)* compared to wild-type N2 worms. In the *oga-1(ok1207)* worms, more sperm were observed. White arrows indicate examples of sperm nuclei. (B) Whereas most of the strains analyzed saw a decrease in the number of sperm nuclei present at day 30 of ARD, *oga-1(ok1207)* had an overall increase compared to wildtype. However, these dynamics did not correlate with changes in brood size among selfed individuals. Green arrows indicate increased brood numbers vs wildtype and red arrows indicate decreased brood vs wildtype. (TIF)

**S4 Fig. 24 hours of ARD results in noticeable decrease in ORO staining.** (A) Decrease in ORO signal were observable at 24 hr after worms were placed on ARD plates (as compared to control fed worms in Fig 5), as shown with representative strains. (B) Strains were stained with DAPI to demonstrate efficient small-molecule penetrance of cuticles across strains, indicating that changes in ORO staining are not related to differences in cuticle penetrance between strains. (TIF)

**S5 Fig. Additional strains stained for ORO after 30 days of ARD.** In addition to the *ogt-1*-dependent pathway, we also looked at strains with diverse phenotypes to see if changes were unique to the *ogt-1* pathway. We observed that outside of the *ogt-1* pathway, strains that did not influence either entry or exit (*age-1(hx546)*), strains that also only influenced entry (*sir-2.1 (ok424)*), and strains that influenced only recovery (*skn-1(zj15)*) had a marked increase in ORO staining/TAG stores compared to both wildtype and the *ogt-1* pathway. We also

observed this same pattern with genes downstream of *nhr-49*, such that *acs-2* (with a defective ARD entry) had reduced ORO staining but *fat-7* (no defect in ARD entry) had a stronger ORO signal. Arrows indicate retained embryos, which show high ORO staining. For easier comparison to control, the image of N2 worms in ARD from Fig 5 is included here.
(TIF)

**S6 Fig. Changes in glycogen and trehalose levels before and after ARD.** (A) Carminic acid staining (indicative of glycogen and trehalose levels) varied greatly between strains. As we have previously reported [22], *ogt-1(1474)* showed slightly higher levels of carminic acid staining than other strains in standard husbandry conditions, though in this study this change did not reach significance. After 30 days of ARD (lower panels) we noted that *nhr-49(nr2041)* had a dramatic increase in staining, while wild type and *ogt-1(ok1474)* did not. These results did not correlate with the observed defect in ARD entry for these strains. (B) ImageJ based quantification of carminic acid fluorescence by pixel intensity. *P*-value **** = <0.0001, as determined by two-way ANOVA.
(TIF)

**S1 Table. Brood size variability.** The table shows the average brood size under normal husbandry (control) or re-fed animals following 5 or 10 days in ARD. A high degree of variability was observed between and within experimental replicates for the average brood size (SEM). These brood sizes did not correlate to fate adoption of each strain nor with influence on any of stages of ARD. Red indicates a decrease compared to wildtype whereas green indicates an increase vs wildtype.
(PDF)

**S1 Methods. Oil Red O image analysis methodology.** Detailed description of image analysis and quantification of images of Oil Red O stained *C. elegans* adults.
(PDF)

# Acknowledgments

The authors wish to thank the members of the Hanover and Krause labs and the Baltimore-Washington Worm Club for helpful discussion and input on this manuscript. We also acknowledge the *C. elegans* Genetics Consortium for the many *C. elegans* strains provided by this resource. We would like to thank Marc Van Gilst for supplying the original *nhr-49* allele used in the study, and to Dr. Todd Lamitina for providing the *ogt-1(dr89[K957M])* allele.

# Author Contributions

**Conceptualization:** Moriah Eustice, Salil Ghosh, Michelle R. Bond, John A. Hanover.

**Data curation:** Moriah Eustice, Daniel Konzman, Salil Ghosh.

**Formal analysis:** Moriah Eustice, Daniel Konzman, Jeff M. Reece, John A. Hanover.

**Funding acquisition:** Andy Golden, John A. Hanover.

**Investigation:** Moriah Eustice.

**Methodology:** Moriah Eustice, Jeff M. Reece, Michelle R. Bond.

**Resources:** Jhullian Alston, Tyler Hansen, Andy Golden.

**Supervision:** John A. Hanover.

**Visualization:** Moriah Eustice.

Writing – **original draft:** Moriah Eustice, Daniel Konzman, John A. Hanover.

Writing – **review & editing:** Moriah Eustice, Daniel Konzman, Michelle R. Bond, Lara K. Abramowitz, John A. Hanover.

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
