## [Decision Letter · Decision Letter 0]

9 May 2022

PONE-D-22-09391Nutrient Sensing Pathways Regulating Adult Reproductive Diapause in C. elegansPLOS ONE

Dear Dr. Hanover

Thank you for submitting your manuscript to PLOS ONE. After careful consideration, we feel that it has merit but does not fully meet PLOS ONE’s publication criteria as it currently stands. Therefore, we invite you to submit a revised version of the manuscript that addresses the points raised during the review process.

Please submit your revised manuscript by Jun 19 2022 11:59PM. If you will need more time than this to complete your revisions, please reply to this message or contact the journal office at plosone@plos.org. Please include the following items when submitting your revised manuscript:A rebuttal letter that responds to each point raised by the academic editor and reviewer(s). You should upload this letter as a separate file labeled 'Response to Reviewers'.A marked-up copy of your manuscript that highlights changes made to the original version. You should upload this as a separate file labeled 'Revised Manuscript with Track Changes'.An unmarked version of your revised paper without tracked changes. You should upload this as a separate file labeled 'Manuscript'.

We look forward to receiving your revised manuscript.

Kind regards,

Myon-Hee Lee, Ph.D

Academic Editor

PLOS ONE

Journal Requirements:

Reviewers' comments:

Reviewer's Responses to Questions

**Comments to the Author**

1. Is the manuscript technically sound, and do the data support the conclusions?

Reviewer #1: Partly

Reviewer #2: Yes

2. Has the statistical analysis been performed appropriately and rigorously? 

Reviewer #1: Yes

Reviewer #2: Yes

3. Have the authors made all data underlying the findings in their manuscript fully available?

Reviewer #1: Yes

Reviewer #2: Yes

4. Is the manuscript presented in an intelligible fashion and written in standard English?

Reviewer #1: Yes

Reviewer #2: Yes

5. Review Comments to the Author

Reviewer #1: Eustice et al present a detailed genetic analysis of the metabolic requirements for a form of caloric restriction in C. elegans called adult reproductive diapause (ARD). From their data, they conclude:

1) Initiation of ARD is regulated by fatty acid metabolism, sirtuins, AMPK, and the OGlcNAc pathway

2) Maintenance of ARD is not influenced by any of the tested metabolic pathways

3) Recovery from ARD required energy sensing, stress response, insulin-like signaling, and TOR pathway function

4) Fatty acid B-oxidation regulates ARD through a ogt-1-nhr-49 pathway

5) Mutants with defects in ARD entry also exhibit changes in the levels of neutral lipids.

Overall, the experiments are extensive and make excellent use of both C. elegans and the ARD pathway to define genetic requirements for this form of caloric restriction. Experiments were performed with a high level of replication and statistical analysis of the data was robust and well done. The discussion is thorough, scholarly, and appropriate. I have some concerns related to the conclusions drawn from the genetic epistasis and the presumed functions of O-GlcNAc pathway and felt that these data could also be interpreted in ways not discussed by the authors.

Major points

1) The authors interpret the ARD entry phenotypes of the nhr-49; ogt-1, nhr-49; oga-1, and acs-2;ogt-1 double mutants in Figure 1 and 4 to mean that these genes function in the same pathway to regulate ARD entry. This is because the double mutants have phenotypes similar to the single mutants. However, these data appear to have a classic ‘floor effect’ problem. For example, the alternative hypothesis is that the genes operate in different pathways. In such a case, the phenotypes should be additive, leading to lower % ARD entry than either of the single mutants. However, the mutants they are analyzing already have extremely low % ARD entry (<10%) and the variability in this rate is fairly large (CV appears to be close to 100% for most). Would it even be possible to detect the alternative hypothesis that the genes function in different pathways? This has significant implications related to the model presented in Figure 6 and in the conclusions made in the discussion.

2) The authors assume that the role of ogt-1 in ARD entry is due to its catalytic addition of O-GlcNAc. However, there is ample evidence that ogt-1 also has non-catalytic roles, particularly in C. elegans. Moreover, the fact that oga-1 mutants exhibit milder phenotypes suggests there may be other aspects to the role of ogt-1 in ARD than O-GlcNAc cycling. The mutant alleles they are using eliminate both catalytic and non-catalytic activities of ogt-1. To better support their conclusion that the role of ogt-1 in ARD relates to the addition of O-GlcNAc, the authors should test catalytically inactive ogt-1 point mutants, which are publicly available.

3) The authors make several qualitative comparisons related to Oil Red O staining that I do not observe. For example, they comment that the ORO staining in ‘frozen’ embryos is stronger in the nhr-49 and ogt-1 single mutants after 30 days of ARD as shown in Figure 5 (I presume they mean compared to N2?). To me, the mutants look either lower than or similar to N2 in these images and no quantification of the embryo ORO levels was performed. What is the evidence for this conclusion? Likewise, Figure S5 lacks an N2 control for comparison. But with the exception of the age-1 mutant, I’m not sure any of the other are obviously ‘stronger’ than the N2 image in Figure S5 without some type of quantification.

4) Minor points

a. I’m not sure I understand the DAPI experiment in Fig S4A. Is this done in live or fixed worms? I think the authors are trying to determine if some of the mutants might have more or less cuticular permeablity to ORO that might explain the differences in staining? But if this is being done in fixed animals (as is the case for ORO), cuticular permeability should not matter since the animal is fully permeabilized.

b. Is the term ‘frozen embryo’ typical for ARD field? I find this to sometimes be confusing, ie is it referring to how the sample was prepared? Would ‘arrested embryo’ serve the same purpose? Seems a little more precise. But I certainly understand sticking with frozen if that is the field standard.

c. Line 176 – I think you mean ‘Hatched L1s’ and not ‘Hatched L4s’?

Reviewer #2: The adult reproductive diapause (ARD) of C. elegans is an intriguing phenomenon from which there is still so much to learn. In this manuscript by Eustine et al, they divide the ARD in three steps: 1) entry, 2) maintenance, and 3) recovery. They explored many pathways that participate in each of these steps. They identified several regulators of ARD initiation like sir-2.1, aak-2, acs-2, ogt-1 and oga-1. They also identified skn-1, rsks-1 and daf-16 as important pathways for ARD recovery. Unexpectedly they did not find any pathway involved in ARD maintenance. They also found that the loss of the O-GlcNAc cycling enzymes converges on the fatty acid metabolic pathways that involved mitochondrial fatty acid B-oxidation. The manuscript contains a fair amount of work that has merits and contributes significantly into the understanding of ARD. However, there are few concerns. The first concern is that the authors did not review carefully all the literature about ARD. The original paper by Angelo and Van Gilst contains two findings that are not reproducible. The first one is that embryos observed during ARD are not arrested. Embryogenesis continues normally during ARD (Seidel and Kimble, 2011). Gametes production does not stop during ARD, in fact oogenesis continues slowly, and that is the main reason there are always embryos in the utero (Carranza-García and Navarro, 2019).

The second one is that Angelo and Van Gilst paper showed that the caspase ced-3 is essential to maintain fertility, however later was shown that ced-3 animals are able to recover their fertility after ARD (Carranza-García and Navarro, 2019).

The insulin pathway was also originally implicated in ARD in 2019. An important contribution to the field was made in 2020 by the group of Antebi showing that the transcription factor HLH-30/TFEB promotes the morphological and physiological remodeling involved in ARD entry, survival and recovery and is not included in the discussion of this manuscript. It would be important to make the appropriate correction throughout the manuscript, and integrate all this literature to strength the paper.

6. PLOS authors have the option to publish the peer review history of their article (what does this mean?). If published, this will include your full peer review and any attached files.

Reviewer #1: **Yes: **Todd Lamitina

Reviewer #2: No

---

## [Author Response · Author response to Decision Letter 0]

21 Jul 2022

Specific responses to reviewer comments can be found in attachment labelled "response to reviewers".

---

## [Decision Letter · Decision Letter 1]

8 Aug 2022

PONE-D-22-09391R1Nutrient Sensing Pathways Regulating Adult Reproductive Diapause in C. elegansPLOS ONE

Dear Dr. Hanover,

Thank you for submitting your manuscript to PLOS ONE. After careful consideration, we feel that it has merit but does not fully meet PLOS ONE’s publication criteria as it currently stands. Therefore, we invite you to submit a revised version of the manuscript that addresses the points raised during the review process.

We look forward to receiving your revised manuscript.

Kind regards,

Myon-Hee Lee, Ph.D

Academic Editor

PLOS ONE

Journal Requirements:

Reviewers' comments:

Reviewer's Responses to Questions

**Comments to the Author**

1. If the authors have adequately addressed your comments raised in a previous round of review and you feel that this manuscript is now acceptable for publication, you may indicate that here to bypass the “Comments to the Author” section, enter your conflict of interest statement in the “Confidential to Editor” section, and submit your "Accept" recommendation.

Reviewer #1: All comments have been addressed

Reviewer #2: All comments have been addressed

2. Is the manuscript technically sound, and do the data support the conclusions?

Reviewer #1: Yes

Reviewer #2: Yes

3. Has the statistical analysis been performed appropriately and rigorously? 

Reviewer #1: Yes

Reviewer #2: Yes

4. Have the authors made all data underlying the findings in their manuscript fully available?

Reviewer #1: Yes

Reviewer #2: Yes

5. Is the manuscript presented in an intelligible fashion and written in standard English?

Reviewer #1: Yes

Reviewer #2: Yes

6. Review Comments to the Author

Reviewer #1: The authors have done an excellent job of responding to my previous comments and suggestions. The lack of an ADR phenotype in the catalytically dead ogt-1 allele is particularly exciting. Outstanding study - congratulations!

Reviewer #2: The authors have responded satisfactorily to all my concerns. The manuscript has improved considerably and will be ready for acceptance after few minor corrections.

1) Authors have changed the term “embryo arrest” in most of the manuscript but there are still a couple of places in which they still use this term. Please change it in lines 307 and 320.

2) Correct the line indent in line 369.

3) Line 505: check partially spelling.

This reviewer suggests outlining the gonad in the ARD worms in Figure 2 because it is not easy to see them.

7. PLOS authors have the option to publish the peer review history of their article (what does this mean?). If published, this will include your full peer review and any attached files.

Reviewer #1: No

Reviewer #2: No

---

## [Author Response · Author response to Decision Letter 1]

11 Aug 2022

Suggested correction/changes to figure 2 have been make

---

## [Decision Letter · Decision Letter 2]

23 Aug 2022

Nutrient Sensing Pathways Regulating Adult Reproductive Diapause in C. elegans

PONE-D-22-09391R2

Dear Dr. Hanover

We’re pleased to inform you that your manuscript has been judged scientifically suitable for publication and will be formally accepted for publication once it meets all outstanding technical requirements.

Kind regards,

Myon Hee Lee, Ph.D

Academic Editor

PLOS ONE

Additional Editor Comments (optional):

Reviewers' comments:

Reviewer's Responses to Questions

**Comments to the Author**

1. If the authors have adequately addressed your comments raised in a previous round of review and you feel that this manuscript is now acceptable for publication, you may indicate that here to bypass the “Comments to the Author” section, enter your conflict of interest statement in the “Confidential to Editor” section, and submit your "Accept" recommendation.

Reviewer #2: All comments have been addressed

2. Is the manuscript technically sound, and do the data support the conclusions?

Reviewer #2: Yes

3. Has the statistical analysis been performed appropriately and rigorously? 

Reviewer #2: Yes

4. Have the authors made all data underlying the findings in their manuscript fully available?

Reviewer #2: Yes

5. Is the manuscript presented in an intelligible fashion and written in standard English?

Reviewer #2: Yes

6. Review Comments to the Author

Reviewer #2: -----------------------------------------------------------------------------------------------------

The authors have responded satisfactorily to my concerns. The paper is ready for publication.

7. PLOS authors have the option to publish the peer review history of their article (what does this mean?). If published, this will include your full peer review and any attached files.

Reviewer #2: No

---

## [Editor Report · Acceptance letter]

7 Sep 2022

PONE-D-22-09391R2 

Nutrient sensing pathways regulating adult reproductive diapause in *C. elegans*

Dear Dr. Hanover:

I'm pleased to inform you that your manuscript has been deemed suitable for publication in PLOS ONE. Congratulations! Your manuscript is now with our production department. 

Kind regards, 

on behalf of

Dr. Myon-Hee Lee 

Academic Editor

PLOS ONE